



# BoundaryLayerDynamics.jl v1.0: a modern codebase for atmospheric boundary-layer simulations

Manuel F. Schmid[1,2], Marco G. Giometto[2], Gregory A. Lawrence[1], and Marc B. Parlange[3,4]

[1]Dept. of Civil Engineering, University of British Columbia, Vancouver, Canada
[2]Dept. of Civil Engineering & Engineering Mechanics, Columbia University, New York, NY, USA
[3]Department of Civil Engineering, Monash University, Melbourne, VIC, Australia
[4]University of Rhode Island, Kingston, RI, USA

**Correspondence:** Manuel F. Schmid (mfs2173@columbia.edu)

**Abstract.** We present BoundaryLayerDynamics.jl, a new code for turbulence-resolving simulations of atmospheric boundary-layer flows as well as canonical turbulent flows in channel geometries. The code performs direct numerical simulation as well as large-eddy simulation using a hybrid (pseudo)spectral and finite-difference approach with explicit time advancement. Written in Julia, the code strives to be flexible and adaptable without sacrificing performance, and extensive automated tests aim to
ensure that the implementation is and remains correct. We show that the simulation results are in agreement with published results and that the performance is on par with an existing Fortran implementation of the same methods.

## 1   Introduction

Since Deardorff's early studies (Deardorff, 1969, 1970a, b), numerical simulations of the three-dimensional, unsteady flow field have become an integral part of microscale atmospheric boundary-layer (ABL) research. Direct numerical simulation
(DNS) provides an extremely accurate tool to study fundamental properties of turbulent flows and their scaling from low to moderate Reynolds numbers (Moin and Mahesh, 1998). Large-eddy simulation (LES) provides a numerical model for a wide range of ABL flow phenomena at realistic Reynolds numbers while relying on modest, well-supported modeling assumptions (Meneveau and Katz, 2000; Stoll et al., 2020). Together, DNS and LES constitute the backbone of the computational study of turbulent flow dynamics and have contributed many insights to our current understanding.

Many different implementation of these methods are in use for ABL research and continue to be actively developed. Projects such as PALM (Maronga et al., 2020), OpenFOAM (Chen et al., 2014), and WRF (Skamarock et al., 2021) develop open-source codes with broad applicability in a community effort. In addition, many research groups have their own codes that have been developed and extended over decades and are passed person-to-person. Some studies also rely on commercial software such as Ansys Fluent, although not having access to implementation details is problematic for scientific reproducibility.

Codes differ along many dimensions, perhaps most importantly in the physical models that are implemented. The numerical methods used to compute a solution can also lead to important differences, particularly for LES, where the smallest resolved scales of motion are an integral part of the turbulence dynamics (Kravchenko and Moin, 1997). The performance characteristics of a code are also central as most turbulence-resolving simulations continue to be limited by their computational cost. Further-



more, there can be important differences in the effort and model-specific experience required for setting up simulations and
making changes and additions to the source code while ensuring the correctness of the results. When developing a new code
or selecting an existing model for a simulation, these different qualities have to be weighed against each other and trade-offs
are inevitable.

The advent of the Julia programming language (Bezanson et al., 2017) represents a shift in the landscape of possible trade-
offs between conflicting goals. Publicly launched in 2012 and stabilized with version 1.0 in 2018, Julia promises to combine
the performance of Fortran, C, and C++ with the convenience of Python, Matlab, and R. Automatic memory management,
dynamic typing with type inference, multiple dispatch, and a built-in modern package manager facilitate rapid development
of clear, concise code that keeps orthogonal functionality separate. At the same time, Julia's "just-ahead-of-time" compilation
model allows code to run with no or minimal computational overhead.

In this paper, we present and discuss BoundaryLayerDynamics.jl, a new code for turbulence-resolving flow simulation
optimized for ABL research. The code has been written to provide core functionality for DNS and LES of channel-flow
configurations with a focus on making it easy to use, adapt, and extend the code without jeopardizing the correctness of the
results. To achieve better trade-offs along these dimensions, the implementation relies on the Julia programming language, on
automated testing, and on a modular design.

The suitability of the new code is of course not limited to simulations of ABL turbulence. In fact, none of the current
functionality is specific to ABL applications. However, the choice of physical models and numerical methods is guided by the
needs of such applications and future developments will similarly prioritize those use cases.

The code solves the incompressible Navier–Stokes equations relying on (pseudo)spectral and finite difference methods for
discretization in the horizontal and vertical directions respectively, making use of the Message Passing Interface (MPI) for
parallelization in the vertical direction, and performing fully explicit time integration. The details of the numerical methods
are described in section 2. The implementation is validated via a number of automated tests as described in section 3, where
we also present a validation against DNS and LES results computed with different codes. The performance analysis presented
in section 4 shows that the computational cost is comparable to a Fortran implementation of the same numerical approach and
that parallel performance scales favorably up to the maximum supported number of parallel processes.

BoundaryLayerDynamics.jl is open source software and available under the MIT License through the official Julia package
repository and on GitHub[1], where the public repository of the package is currently hosted. The version described in this article
is archived on Zenodo (Schmid, 2023).

## 2   Governing equations and numerical methods

The choice of governing equations and numerical methods is guided by the goal of studying the turbulent flow dynamics in the
atmospheric boundary layer. Other turbulent flows in engineering and in the natural environment are also considered insofar as
their requirements do not conflict with those of atmospheric boundary-layer flows.

---

[1]https://github.com/efpl-columbia/BoundaryLayerDynamics.jl





It is well-established that the Navier–Stokes equations are an extremely accurate physical model for flows with a Knudsen number $Kn \ll 1$ and that compressibility effects are minimal for flows with a Mach number $Ma \lesssim 1/3$ (Panton, 2013). Since these conditions are met for atmospheric boundary-layer flows, the incompressible Navier–Stokes equations,

$$\frac{\partial u_i}{\partial t} + u_j \left( \frac{\partial u_i}{\partial x_j} - \frac{\partial u_j}{\partial x_i} \right) = \frac{1}{Re} \frac{\partial u_i}{\partial x_j} x_j - \frac{\partial p}{\partial x_i} + f_i \quad \text{and} \quad \frac{\partial u_i}{\partial x_i} = 0, \tag{1}$$

are used as the mathematical model for this work, given here with the rotational form of the advection term (Orszag, 1971a), for which the total kinematic pressure $p = p^{\text{gauge}}/\rho + \frac{1}{2} u_i u_i$ includes both the static and dynamic pressure. Quantities are non-dimensionalized with a length scale $\mathcal{L}$ and a velocity $\mathcal{U}$, producing the Reynolds number $Re = \mathcal{U}\mathcal{L}/\nu$. The Cartesian coordinates $x_i$ denote the primary horizontal ($i = 1$, usually streamwise), secondary horizontal ($i = 2$, usually cross-stream), and vertical ($i = 3$) directions while the corresponding velocity components are given as $u_i$. The term $f_i$ denotes the components 65   of a body force, usually gravity or a constant pressure gradient, and the kinematic viscosity $\nu$ and fluid density $\rho$ are assumed scalar constants.

While this formulation can serve as a reasonable representation of a neutrally stratified atmospheric boundary layer, there are many important ABL processes that are not included. Coriolis forces, temperature, and humidity in particular are of central importance for many applications. The current code is meant to provide the core functionality necessary for ABL flow 70   simulations and serve as a foundation for a more comprehensive set of physical and numerical models that can be added over time.

For flows with a moderate to high Reynolds number, current computational capabilities generally do not permit resolving the full range of scale of motions. In this case, the filtered Navier–Stokes equations,

$$\frac{\partial \widetilde{u}_i}{\partial t} + \widetilde{u}_j \left( \frac{\partial \widetilde{u}_i}{\partial x_j} - \frac{\partial \widetilde{u}_j}{\partial x_i} \right) + \frac{\partial \tau_{ij}^{\text{sgs}}}{\partial x_j} = \frac{1}{Re} \frac{\partial \widetilde{u}_i}{\partial x_j \partial x_j} - \frac{\partial \widetilde{p}}{\partial x_i} + \widetilde{f}_i \quad \text{and} \quad \frac{\partial \widetilde{u}_i}{\partial x_i} = 0, \tag{2}$$

are used as the computational model, where $\widetilde{u}_i$ represents the spatially filtered velocity field, i.e. $\widetilde{u}_i = \int G(\boldsymbol{r}, \boldsymbol{x}) u_i(\boldsymbol{x} - \boldsymbol{r}) \mathrm{d}\boldsymbol{r}$ with $G$ defining the filtering operation. The subgrid-scale stress tensor $\tau_{ij}^{\text{sgs}} = \tau_{ij}^{\text{R}} - \frac{1}{3} \tau_{ii}^{\text{R}} \delta_{ij}$ represents the anisotropic component of the residual stress tensor $\tau_{ij}^{\text{R}} = \widetilde{u_i u_j} - \widetilde{u}_i \widetilde{u}_j$ and has to be modeled as a function of the resolved velocity $\widetilde{u}_i$. The modified pressure $\widetilde{p} = \widetilde{p}^{\text{gauge}}/\rho + \frac{1}{2} \widetilde{u}_i \widetilde{u}_i + \frac{1}{3} \tau_{ii}^{\text{R}}$ now includes contributions from the filtered gauge pressure $\widetilde{p}^{\text{gauge}}$, the resolved kinetic energy, and the unresolved kinetic energy. The forcing term $\widetilde{f}_i$ is simply the spatially filtered $f_i$. In the following, the same 80   notation is used to represent both the unfiltered (DNS) and filtered (LES) equations to simplify the notation.

The past decades have seen several efforts to develop a suitable model for $\tau_{ij}^{\text{sgs}}$ (Smagorinsky, 1963; Schumann, 1975; Bardina et al., 1980; Germano et al., 1991; Meneveau et al., 1996; Porté-Agel et al., 2000; Bou-Zeid et al., 2005). The current implementation includes the static Smagorinsky (1963) subgrid-scale model

$$\tau_{ij}^{\text{sgs}} \approx -2 l_{\text{S}}^2 \mathcal{S} S_{ij}, \tag{3}$$

where $S_{ij} = 1/2 \left( \frac{\partial u_i}{\partial x_j} + \frac{\partial u_j}{\partial x_i} \right)$ is the resolved strain rate, $\mathcal{S} = \sqrt{2 S_{ij} S_{ij}}$ is the characteristic or total strain rate, and $l_{\text{S}}$ is the Smagorinsky lengthscale, taken to be the product of the filter width $\Delta$ and a constant Smagorinsky coefficient $C_{\text{S}}$.



Specifying boundary conditions for turbulent flows remains a challenging research problem, since chaotic velocity fluctuations have to be prescribed in a physically accurate manner. To avoid these difficulties, turbulence-resolving simulations are often run with periodic boundary conditions. While this requires that the problem is formulated such that it can be approx-
imated with a periodic flow field, unphysical border regions are avoided and accurate results can be obtained in the whole domain as long as the domain is large enough to accommodate all relevant scales of motion. Furthermore, the flow field can then be expressed in terms of periodic basis functions. For a horizontally-periodic domain of size $L_1 \times L_2$, the velocity field can be written as

$$u_i(x_1, x_2, x_3) = \sum_{-\infty < \kappa_1 < \infty} \sum_{-\infty < \kappa_2 < \infty} \hat{u}_i^{\kappa_1 \kappa_2}(x_3) \, \mathrm{e}^{i\kappa_1 2\pi x_1/L_1} \, \mathrm{e}^{i\kappa_2 2\pi x_2/L_2} \,. \tag{4}$$

With similar expressions for the pressure $p$ and the forcing $f_i$, we can rewrite the governing equations for a single mode with wavenumber $\kappa_1$ in $x_1$-direction and $\kappa_2$ in $x_2$-direction,

$$\frac{\partial}{\partial t} \hat{u}_i^{\kappa_1 \kappa_2}(x_3) + \sum_{-\infty < \kappa_1' < \infty} \sum_{-\infty < \kappa_2' < \infty} \left( \hat{D}_j^{\kappa_1' \kappa_2'} \hat{u}_i^{\kappa_1' \kappa_2'}(x_3) - \hat{D}_i^{\kappa_1' \kappa_2'} \hat{u}_j^{\kappa_1' \kappa_2'}(x_3) \right) \hat{u}_j^{(\kappa_1 - \kappa_1')(\kappa_2 - \kappa_2')}(x_3)$$

$$+ \hat{D}_j^{\kappa_1 \kappa_2} \hat{\tau}_{ij}^{\mathrm{sgs}\, \kappa_1 \kappa_2}(x_3) = \frac{1}{Re} \left( \hat{D}_j^{\kappa_1 \kappa_2} \right)^2 \hat{u}_i^{\kappa_1 \kappa_2}(x_3) - \hat{D}_i^{\kappa_1 \kappa_2} \hat{p}^{\kappa_1 \kappa_2}(x_3) + \hat{f}_i^{\kappa_1 \kappa_2}(x_3) \,, \tag{5}$$

where $\hat{D}_1^{\kappa_1 \kappa_2} = \frac{2\pi i \kappa_1}{L_1}$, $\hat{D}_2^{\kappa_1 \kappa_2} = \frac{2\pi i \kappa_2}{L_2}$, and $\hat{D}_3^{\kappa_1 \kappa_2} = \frac{\partial}{\partial x_3}$ are the differential operators, with $i$ denoting the imaginary unit. For direct numerical simulations the subgrid-scale term is omitted. The continuity equation becomes

$\hat{D}_i^{\kappa_1 \kappa_2} \hat{u}_i^{\kappa_1 \kappa_2}(x_3) = 0 \,. \tag{6}$

In the vertical direction, turbulence is not homogeneous for ABL flows and periodic boundary conditions are not applicable. For a channel geometry, boundary conditions have to be specified for $u_i(x_3 = 0)$ and $u_i(x_3 = L_3)$ with the constraint that

$$\int_0^{L_2} \int_0^{L_1} u_3(x_3 = 0) - u_3(x_3 = L_3) \, \mathrm{d}x_1 \mathrm{d}x_2 = 0 \,, \tag{7}$$

which can be obtained from integrating the continuity equation over the whole domain. A number of engineering flows such as
smooth-wall open and closed channel flows can be modeled with Dirichlet and Neumann boundary conditions. The complex boundaries of atmospheric flows can require significant modeling effort and simulations generally have to partially resolve surfaces (e.g. immersed boundary method, terrain-following coordinates) or represent their effect with a wall model for $\tau_{i3}^{\mathrm{sgs}}$, usually formulated for the discretized equations (Piomelli and Balaras, 2002). The current implementation includes an algebraic equilibrium rough-wall model defined similar to Mason and Callen (1986) with

$$\tau_{i3}^{\mathrm{sgs}}(x_3 = 0) \approx \frac{-\kappa^2 \sqrt{u_1(x_3^{\mathrm{ref}})^2 + u_2(x_3^{\mathrm{ref}})^2}}{\log(x_3^{\mathrm{ref}}/z_0)^2} u_i(x_3^{\mathrm{ref}}) \tag{8}$$

for $i = 1, 2$ and $\tau_{33}^{\mathrm{sgs}}(x_3 = 0) = 0$, where $z_0$ is the roughness length, $\kappa \approx 0.4$ is the von Kármán constant, and $x_3^{\mathrm{ref}} > z_0$ is a reference height at which the (resolved) velocity is obtained, usually chosen as the first grid point. To improve the near-wall



behavior of the subgrid-scale model, the Smagorinsky length scale is adjusted to $l_\mathrm{S}^{-n} = (C_\mathrm{S}\Delta)^{-n} + (\kappa x_3)^{-n}$ as proposed by Mason and Thomson (1992), with $n = 2$ as the default value.

For the numerical solution of equations (5) and (6) we limit ourselves to $N_1 \times N_2$ wavenumbers at $N_3$ vertical grid points. The wavenumbers are selected symmetrically around $\kappa_i = 0$, i.e. $|\kappa_1| \leq (N_1 - 1)/2$ and $|\kappa_2| \leq (N_2 - 1)/2$. This results in an odd number of wavenumbers in each direction and avoids the need for a special treatment of Nyquist frequencies. Since $\hat{\phi}^{-\kappa_1 - \kappa_2} = \hat{\phi}^{\kappa_1 \kappa_2 *}$ for any real-valued $\phi$, we only need to explicitly solve for half the modes and can obtain the others through complex conjugation. In vertical direction, equidistant grid points are selected from the interval $[0,1]$, which is then mapped

to the domain with a function $x_3 : [0,1] \to [0, L_3]$, $\zeta \mapsto x_3(\zeta)$. This function can be used for grid stretching in the vertical direction; the choice of $x_3 : \zeta \mapsto L_3\zeta$ defines a uniform grid. A staggered arrangement of grid points with $\zeta_C$ at the center of the $N_3$ segments and $\zeta_I$ at the $N_3 - 1$ interfaces between them, i.e.,

$$
\begin{aligned}
\zeta_C \in \left\{ \frac{1/2}{N_3}, \frac{3/2}{N_3}, \ldots, \frac{N_3 - 1/2}{N_3} \right\} &\qquad \text{for} \quad u_1, u_2, p, f_1, f_2, \tau_{ii}^{\mathrm{sgs}}, \tau_{12}^{\mathrm{sgs}}, \\
\zeta_I \in \left\{ \frac{1}{N_3}, \frac{2}{N_3}, \ldots, \frac{N_3 - 1}{N_3} \right\} &\qquad \text{for} \quad u_3, f_3, \tau_{13}^{\mathrm{sgs}}, \tau_{23}^{\mathrm{sgs}},
\end{aligned}
\tag{9}
$$

avoids the need to specify boundary conditions for the pressure field, prevents odd–even decoupling, and results in a smaller

effective grid spacing (Ferziger et al., 2020). When running large-eddy simulations, the discretization implicitly defines the spatial filter $G$ and the filter width $\Delta$ is taken to be $\Delta = \sqrt[3]{\Delta_1 \Delta_2 \Delta_3}$ (Scotti et al., 1993) with $\Delta_1 = L_1/N_1$, $\Delta_2 = L_2/N_2$, and $\Delta_3 = 1/N_3 \mathrm{d}x_3/\mathrm{d}\zeta$.

    The horizontal derivatives $\hat{D}_1^{\kappa_1 \kappa_2}$ and $\hat{D}_2^{\kappa_1 \kappa_2}$ can be computed exactly. For the vertical derivative $\hat{D}_3^{\kappa_1 \kappa_2}$, we use central second-order finite differences on the staggered $\zeta$-nodes (Moin and Verzicco, 2016) as well as the analytical derivative of

$x_3(\zeta)$, i.e.

$$
\hat{D}_3^{\kappa_1 \kappa_2} \hat{\phi}^{\kappa_1 \kappa_2} \Big|_\zeta = \frac{\mathrm{d}\zeta}{\mathrm{d}x_3}\Big|_\zeta \frac{\partial \hat{\phi}^{\kappa_1 \kappa_2}}{\partial \zeta}\Big|_\zeta \approx \frac{\mathrm{d}\zeta}{\mathrm{d}x_3}\Big|_\zeta \frac{\hat{\phi}^{\kappa_1 \kappa_2}(\zeta + \delta\zeta/2) - \hat{\phi}^{\kappa_1 \kappa_2}(\zeta - \delta\zeta/2)}{\delta\zeta}
\tag{10}
$$

for any field $\phi$, where $\delta\zeta \equiv 1/N_3$ is the grid spacing in the $\zeta$ coordinate. Vertical derivatives are therefore evaluated at the opposite set of grid points to the ones where $\phi$ is defined, as typical for staggered grids. At the boundary, one-sided second-order stencils are employed. This approximation of the vertical derivatives results in a truncation error of order $\mathcal{O}(\delta\zeta^2)$.

The non-linear term of Eq. (5) requires further approximations. First, some of the terms (e.g. $i = 1, j = 3$) are evaluated at the opposite set of vertical grid points than where they are required and have to be interpolated. With the simple interpolation $\hat{\phi}(\zeta) \approx 1/2(\hat{\phi}(\zeta - \delta\zeta/2) + \hat{\phi}(\zeta + \delta\zeta/2))$, the truncation error generally increases but remains of order $\mathcal{O}(\delta\zeta^2)$. Furthermore, the double sum can only be computed over the resolved range of wavenumbers, i.e. $|\kappa_1'| \leq (N_1 - 1)/2$ and $|\kappa_2'| \leq (N_2 - 1)/2$, producing another truncation error that decreases exponentially with the number of resolved wavenumbers. The same applies

to the non-linear expressions involved in the evaluation of $\tau_{ij}^{\mathrm{sgs}}$.

    To simplify the computation of non-linear terms and avoid evaluating expensive convolutions, those terms are computed on $N_1^{\mathrm{PD}} \times N_2^{\mathrm{PD}}$ equidistant grid points in the physical domain, relying on the fast Fourier transform (FFT) algorithm for forward and backward transforms (Orszag, 1969, 1971b). In principle, $N_1^{\mathrm{PD}}$ and $N_2^{\mathrm{PD}}$ are parameters that can be chosen independently



of $N_1$ and $N_2$, but the choice of $N_i^{\mathrm{PD}} \geq 1 + 3\kappa_i^{\max}$ avoids introducing aliasing errors for a simple product of two variables
such as the resolved advection term (Patterson and Orszag, 1971). In this case, the physical-domain evaluation is equivalent
to a true spectral Galerkin method computing the convolution of Eq. (5) over all resolved wavenumbers. Contributions from
wavenumbers $|\kappa_i| > (N_i - 1)/2$ are discarded upon return to the Fourier domain and vertical derivatives can be computed
before or after the horizontal Fourier transforms as the two operations commute.

For the more complex non-linear expressions introduced by the SGS model, full dealiasing is generally not feasible and
physical-domain evaluations incur aliasing errors in addition to the truncation errors. This approach, dubbed the pseudospectral
method by Orszag (1971b), can achieve similar accuracy to a Galerkin method (Orszag, 1972). While it is common to set
$N_i^{\mathrm{PD}} = N_i$ for pseudospectral approximations and only discard the Nyquist wavenumber, $N_i^{\mathrm{PD}}$ can in principle be chosen
freely for more control over truncation errors. Furthermore, it can be beneficial to choose different values of $N_i^{\mathrm{PD}}$ for each
non-linear term, since computing the resolved advection requires only nine transforms and is known to be sensitive to aliasing
errors (Kravchenko and Moin, 1997; Margairaz et al., 2018) while the evaluation of the SGS model requires 15 transforms.

Combining the velocity components into a single vector $\hat{\boldsymbol{u}}$ of length $N_1 \times N_2 \times (3N_3 - 1)$, the spatially discretized momentum
equation can be written as

$$\frac{\mathrm{d}\hat{\boldsymbol{u}}}{\mathrm{d}t} \approx \mathrm{Adv}(\hat{\boldsymbol{u}}) + \frac{1}{Re}\Delta\hat{\boldsymbol{u}} + \frac{1}{Re}\hat{\boldsymbol{b}}_\Delta - \mathrm{G}\hat{\boldsymbol{p}} + \hat{\boldsymbol{f}}. \tag{11}$$

Here, $\mathrm{Adv}(\cdot)$ is the non-linear advection operator that includes both resolved and subgrid-scale contributions. $\Delta$ is the linear
operator that computes the Laplacian of each velocity component, with $\hat{\boldsymbol{b}}_\Delta$ denoting the contributions from vertical boundary
conditions. G is the linear operator that computes the gradient of the pressure, discretized as a vector $\hat{\boldsymbol{p}}$ of size $N_1 \times N_2 \times N_3$.
The forcing term $\hat{\boldsymbol{f}}$, a vector of the same length as $\hat{\boldsymbol{u}}$, can be constant in time and space (e.g. pressure-driven channel flow), vary
in time only (e.g. constant-mass-flux channel flow), vary in space only (e.g. baroclinic flow), or even vary in time and space
as a function of $\hat{\boldsymbol{u}}$, which can be used to model vegetation drag or to represent complex geometry with an immersed-boundary
method. The discrete continuity equation,

$$\mathrm{D}\hat{\boldsymbol{u}} + \hat{\boldsymbol{b}}_{\mathrm{D}} \approx 0, \tag{12}$$

contains the linear divergence operator D with the contributions $\hat{\boldsymbol{b}}_{\mathrm{D}}$ from the vertical boundary conditions of $u_3$.

This hybrid approach, relying on spectral approximations in horizontal direction (pseudospectral for the evaluation of $\tau_{ij}^{\mathrm{sgs}}$)
and second-order-accurate finite differences in vertical direction, has long been employed for computational studies of turbu-
lent flows in channel geometries (Moin and Kim, 1982; Moeng, 1984; Albertson and Parlange, 1999a, b). It combines the fast
convergence and low dissipation of spectral methods (Giacomini and Giometto, 2021) with the ease of parallelization and sim-
ple handling of boundary conditions of finite differences. Conversely, handling complex domains and non-periodic boundaries
can be problematic, though still possible (Chester et al., 2007; Schmid, 2015; Li et al., 2016).

Following Perot (1993), we obtain the expressions for time integration through a block LU decomposition of the fully-
discretized equations. This results in expressions in the style of the fractional step method (Chorin, 1968; Temam, 1969), but
avoids the need for boundary conditions for the intermediate velocity and the pressure on a staggered grid and can easily be
adapted when new terms are included or different numerical methods are employed.





Adams–Bashforth methods solving ordinary differential equations of the form $\mathrm{d}\boldsymbol{u}/\mathrm{d}t = f(\boldsymbol{u})$, $\boldsymbol{u}(t_0) = \boldsymbol{u}_0$ can be written as $\boldsymbol{u}^{(n+1)} = \boldsymbol{u}^{(n)} + \Delta t \sum_{i=0}^{s-1} \beta_i f(\boldsymbol{u}^{(n-i)})$, where $\beta_i$ are the coefficients of the method, $s$ is the order of accuracy, and superscripts

denote the time step (Hairer et al., 1993). With $\beta_0 = 1$ this corresponds to the forward Euler method ($s = 1$) while $\beta_0 = 3/2, \beta_1 = -1/2$ gives second-order accuracy ($s = 2$). Applied to the momentum equation (12), this can be written as

$$\hat{\boldsymbol{u}}^{(n+1)} = \hat{\boldsymbol{u}}^{(n)} + \Delta t \sum_{i=0}^{s-1} \beta_i \left( \mathrm{F}(\hat{\boldsymbol{u}}^{(n-i)}) - \mathrm{G}\,\hat{\boldsymbol{p}}^{(n-i)} \right), \tag{13}$$

where terms are grouped with the definition $\mathrm{F}(\hat{\boldsymbol{u}}) \equiv \mathrm{Adv}(\hat{\boldsymbol{u}}) + \frac{1}{Re}\Delta\hat{\boldsymbol{u}} + \frac{1}{Re}\hat{\boldsymbol{b}}_\Delta + \hat{\boldsymbol{f}}$ to simplify the notation. Together with the continuity equation (12), the fully-discretized equations become


$$\hat{\boldsymbol{u}}^{(n+1)} + \mathrm{G}\,\hat{\boldsymbol{\varphi}}^{(n+1)} = \hat{\boldsymbol{u}}^{(n)} + \Delta t \sum_{i=0}^{s-1} \beta_i \,\mathrm{F}\left( \hat{\boldsymbol{u}}^{(n-i)} \right) \quad \text{and}$$
$$\mathrm{D}\,\hat{\boldsymbol{u}}^{(n+1)} = -\hat{\boldsymbol{b}}_{\mathrm{D}} \tag{14}$$

if we group the pressure contributions with $\hat{\boldsymbol{\varphi}}^{(n+1)} \equiv \Delta t \sum_{i=0}^{s-1} \beta_i \hat{\boldsymbol{p}}^{(n-i)}$.

Similarly, explicit $s$-stage Runge–Kutta methods can be written as $\boldsymbol{u}^{(n,i)} = \sum_{k=0}^{i-1} \left( \alpha_{ik}\boldsymbol{u}^{(n,k)} + \Delta t\beta_{ik} f(\boldsymbol{u}^{(n,k)}) \right)$ for $i = 1, \dots, s$, with $\boldsymbol{u}^{(n,0)} = \boldsymbol{u}^{(n)}$ and $\boldsymbol{u}^{(n+1)} = \boldsymbol{u}^{(n,s)}$ (Gottlieb et al., 2009). This is referred to as the Shu–Osher form (Shu and Osher, 1988), where the coefficients $\alpha_{ik}$ and $\beta_{ik}$ are not uniquely determined by the Butcher tableau of the method and can be

chosen to minimize storage requirements. In this case, the fully discretized equations can be written as

$$\hat{\boldsymbol{u}}^{(n,i)} + \mathrm{G}\,\hat{\boldsymbol{\varphi}}^{(n,i)} = \sum_{k=0}^{i-1} \left( \alpha_{ik}\hat{\boldsymbol{u}}^{(n,k)} + \Delta t\beta_{ik}\,\mathrm{F}\left( \hat{\boldsymbol{u}}^{(n,k)}, t^{(n,k)} \right) \right) \quad \text{and}$$
$$\mathrm{D}\,\hat{\boldsymbol{u}}^{(n,i)} = -\hat{\boldsymbol{b}}_{\mathrm{D}} \tag{15}$$

with the definition $\hat{\boldsymbol{\varphi}}^{(n,i)} \equiv \Delta t \sum_{k=0}^{i-1} \beta_{ik} \hat{\boldsymbol{p}}^{(n,k)}$.

Each step or stage requires the solution of a system of equations in the form $\hat{\boldsymbol{u}} + \mathrm{G}\,\hat{\boldsymbol{\varphi}} = \hat{\boldsymbol{a}}$ and $\mathrm{D}\,\hat{\boldsymbol{u}} = \hat{\boldsymbol{b}}$. This can be solved with the LU-decomposition


$$\begin{pmatrix} \mathrm{I} & \mathrm{G} \\ \mathrm{D} & 0 \end{pmatrix} \begin{pmatrix} \hat{\boldsymbol{u}} \\ \hat{\boldsymbol{\varphi}} \end{pmatrix} = \begin{pmatrix} \mathrm{I} & 0 \\ \mathrm{D} & -\mathrm{D}\,\mathrm{G} \end{pmatrix} \begin{pmatrix} \mathrm{I} & \mathrm{G} \\ 0 & \mathrm{I} \end{pmatrix} \begin{pmatrix} \hat{\boldsymbol{u}} \\ \hat{\boldsymbol{\varphi}} \end{pmatrix} = \begin{pmatrix} \mathrm{I} & 0 \\ \mathrm{D} & -\mathrm{D}\,\mathrm{G} \end{pmatrix} \begin{pmatrix} \hat{\boldsymbol{u}}^\star \\ \hat{\boldsymbol{\varphi}} \end{pmatrix} = \begin{pmatrix} \hat{\boldsymbol{a}} \\ \hat{\boldsymbol{b}} \end{pmatrix}, \tag{16}$$

where $\hat{\boldsymbol{u}}^\star \equiv \hat{\boldsymbol{u}} + \mathrm{G}\,\hat{\boldsymbol{\varphi}}$ has been introduced. The steps to compute the solution are therefore

$$\hat{\boldsymbol{u}}^\star = \hat{\boldsymbol{a}},$$
$$\mathrm{D}\,\mathrm{G}\,\hat{\boldsymbol{\varphi}} = \mathrm{D}\,\hat{\boldsymbol{u}}^\star - \hat{\boldsymbol{b}}, \quad \text{and} \tag{17}$$
$$\hat{\boldsymbol{u}} = \hat{\boldsymbol{u}}^\star - \mathrm{G}\,\hat{\boldsymbol{\varphi}},$$

where the second step requires solving a linear system. Since the operator $(\mathrm{D}\,\mathrm{G})$ has no coupling between different wavenumbers, Eq. (17) can be decomposed into $N_1 \times N_2$ tridiagonal systems of size $N_3$. For $\kappa_1 = \kappa_2 = 0$ the system is singular due to



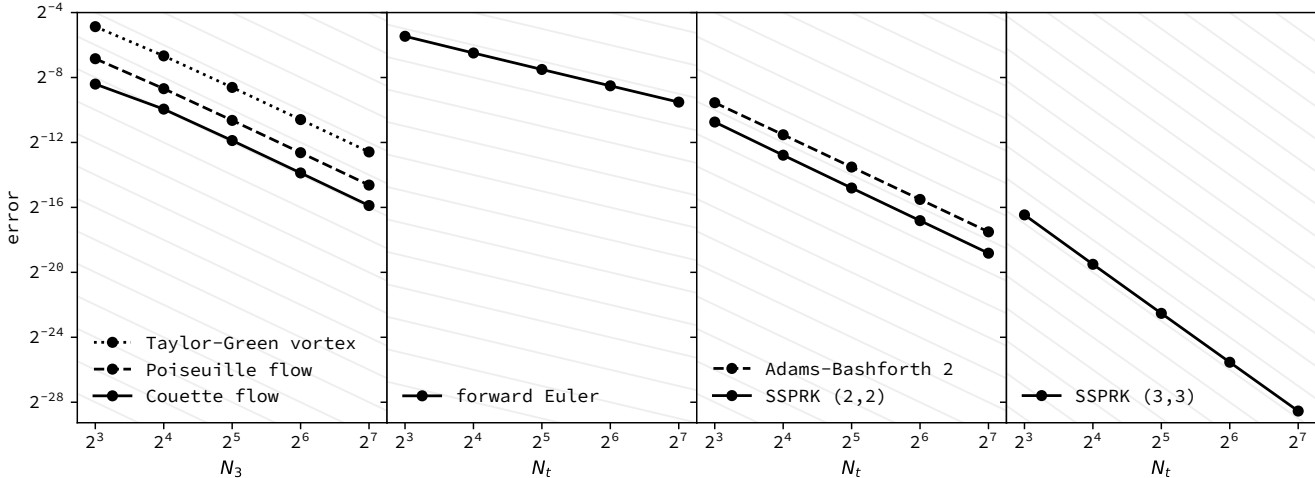

**Figure 1.** Error convergence for transient two-dimensional laminar flows. The left panel shows second-order convergence as the vertical grid resolution is refined for flows set up along the vertical direction and a randomly chosen horizontal direction. The other panels show first-, second-, and third-order convergence as the time resolution is refined for a Taylor–Green vortex set up along the two horizontal directions, in which case the spatial discretization is exact and the order of convergence of the time integration methods is measured. Grid lines show the formal order of convergence for each case.

the fact that the governing equations only include the gradient of the pressure and do not place any restrictions on the absolute magnitude of the pressure variable. This system therefore has to be solved iteratively or the equations have to be regularized, e.g. by specifying an arbitrary value for one element of $\hat{\varphi}$. The current implementation relies on the Thomas algorithm (Quarteroni et al., 2007) to solve the tridiagonal systems and includes the forward Euler and second-order Adams–Bashforth methods for time integration as well as the strong stability preserving Runge–Kutta methods SSPRK (2,2) and SSPRK (3,3) (Gottlieb et al., 2009). Adding other explicit methods is straightforward, provided they can be formulated in a similar fashion.

The simulation code is written in the Julia programming language, relying on the Julia bindings to the FFTW library (Frigo and Johnson, 2005) for fast Fourier transforms. For parallelization, the domain is vertically split into up to $N_3$ blocks that are computed by separate processes exchanging information through the Message Passing Interface (MPI).

## 3  Model validation

The validation efforts presented in this section aim to confirm that the numerical methods are implemented faithfully and that these methods produce physically relevant results. To maintain this confidence as the code is inevitably modified, a focus is placed on automated tests that can be rerun after every change. A set of automated unit tests verifies the expected order of accuracy when computing individual terms of the discretized equations for prescribed velocity fields and when applying the time-integration algorithms to ordinary differential equations. A set of automated integration tests verifies that the solution to



canonical transient two-dimensional laminar flows can be simulated with the expected order of accuracy. Finally, fully turbulent flow solutions are computed and compared to published results produced with different codes. These tests are not automated since they require significant computational resources and have no analytical solution to compare against so there is some degree of judgment required to evaluate the quality of the solution.

The automated tests of individual terms make use of the fact that the implemented numerical methods are exact for certain

velocity fields. The diffusion term and the pressure solver are exact for a function that is the product of truncated Fourier series along horizontal dimensions and a quadratic polynomial in vertical direction. The advection term is only exact for a linear function in vertical direction due to linear interpolations, although the term is still second-order accurate. By constructing such a function with randomized parameters, each term can be computed numerically as well as analytically and matching values give a high degree of confidence in the correctness of the implementation. Furthermore, we can verify the order of

convergence when computing the terms at different grid resolutions for a velocity field that cannot be handled exactly by the implemented methods. The time integration methods are verified in a similar way by solving ordinary differential equations that have analytical solutions with different time steps. We also verify that the tridiagonal solver is exact for random inputs.

To test the full solver including time integration, the automated tests include a number of laminar flow problems, currently the transient Poiseuille and Couette flows as well as decaying Taylor–Green vortices. Numerical solutions computed at different

resolutions are then compared to the analytical solution to ensure that the order of convergence corresponds to formal order of the numerical methods, as shown in Fig. 1. Dimensional parameters such as domain sizes and velocity scales are again chosen randomly since parameters that are zero or unity can mask errors in the solution. For Poiseuille and Couette flows, this includes the horizontal direction of the flow. The two-dimensional Taylor–Green vortices are oriented both in horizontal and vertical planes. For the former, the spatial discretization is exact so the test case verifies the order of convergence of the time integration

method.

The automated tests can be re-run whenever changes are made. By default, tests are run in single-process (serial) mode as well as in multi-process (parallel) mode and multi-process tests are run both with a vertical resolution greater than and equal to the number of processes since those configurations sometimes rely on different code paths. The tests can be run on consumer hardware used for code development, although it is recommended to have at least four CPU cores as some MPI

implementations struggle when cores are oversubscribed.

Turbulence-resolving flow simulations require substantially more computation and are therefore not included in the automated tests that are meant to be run routinely during code development. The validation cases presented below are chosen such that they represent scientifically relevant flow systems while keeping the computational cost moderate. Each case can be simulated in about two hours using 32 MPI processes on a single compute node of the Intel Skylake generation.

The results of direct numerical simulations are not supposed to depend on the exact method used for modeling the flow, at least for lower-order flow statistics. For relatively low Reynolds numbers, simulations have been run with many different codes and with a wide range of parameters such as domain sizes, aspect ratios, and grid resolutions, so the expected simulation results are well-established and have been validated against wind-tunnel measurements (Kim et al., 1987; del Álamo and Jiménez, 2003; Lee and Moser, 2015).





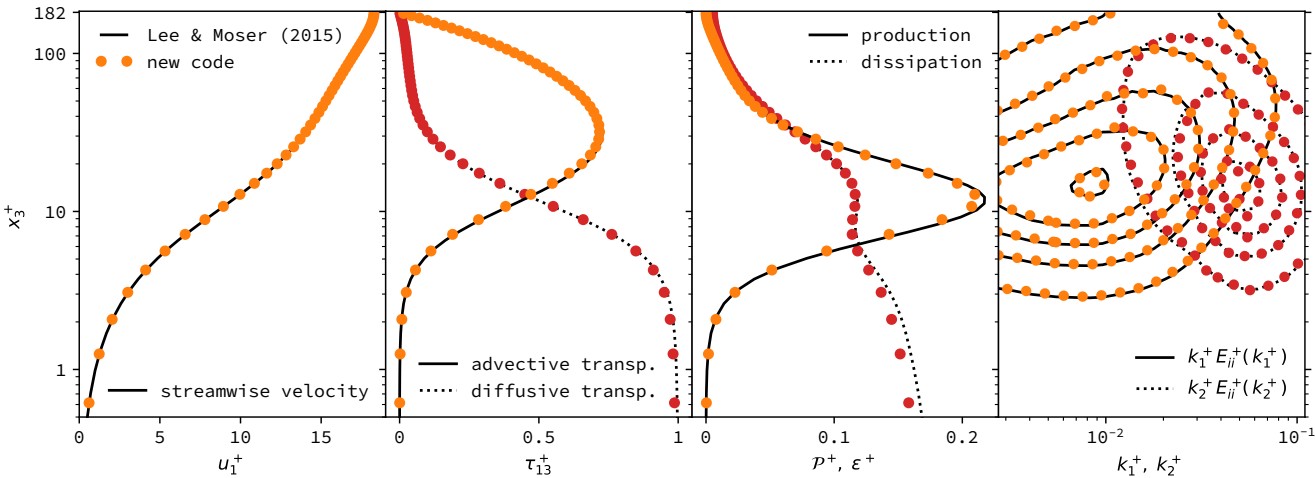

**Figure 2.** Direct numerical simulation (DNS) of a turbulent channel flow at $Re_\tau \approx 180$, validated against data published by Lee and Moser (2015). Mean profiles are shown for the streamwise velocity $u_1^+$, the advective transport $u_1^+ u_3^+$, the diffusive transport $\partial u_1^+/\partial x_3^+$, as well as the production $\mathcal{P}^+$ and (pseudo)dissipation $\varepsilon^+$ of turbulent kinetic energy. The last panel shows contours of the premultiplied turbulent kinetic energy spectra $E_{ii}^+$ along the streamwise ($k_1^+ = 2\pi\kappa_1/L_1^+$) and cross-stream ($k_2^+ = 2\pi\kappa_2/L_2^+$) direction. The superscript $^+$ marks values in inner units, i.e. non-dimensionalized with the friction velocity $u_\tau$ and the kinematic viscosity $\nu$.

In Fig. 2, we show a comparison of a closed-channel flow at $Re_\tau \approx 180$ with data published by Lee and Moser (2015). The friction Reynolds number $Re_\tau = u_\tau \delta/\nu$ is based on the half-channel height $\delta$ and the friction velocity $u_\tau^2 = \nu \frac{\partial u_1}{\partial x_3}\big|_{x_3=0}$ here. The simulation is run with a bulk Reynolds number of $Re_b = U_b\delta/\nu = 20000/7$, where the vertically averaged bulk velocity $U_b$ is held constant by a dynamically adjusted pressure forcing. The solution is computed in a domain of size $4\pi\delta$ in streamwise and $2\pi\delta$ in cross-stream direction. The velocity field is discretized with $255 \times 191$ Fourier modes at 96 vertical grid points that

are spaced according to a sinusoidal grid transform $x_3(\zeta) = \delta + \delta \sin\left((2\zeta - 1)\eta\pi/2\right)/\sin(\eta\pi/2)$ with $\eta = 0.97$. The mean statistics computed over $\sim$17.5 large-eddy turnover times $T_\tau = \delta/u_\tau$ after a spin-up time of $\sim$3.5$T_\tau$ closely match the results from Lee and Moser (2015).

For large-eddy simulation, validation is not as straightforward since results remain relatively sensitive to differences in the modeling approach and in the grid resolution. To validate the new implementation, we limit ourselves to a comparison with a

pre-existing Fortran implementation of the same physical and numerical models (Giometto et al., 2017) and refer to previous publications for validation studies and discussions of limitations of the modeling approach (Porté-Agel et al., 2000; Yue et al., 2007, 2008; Giometto et al., 2016).

In Fig. 3 we show the results of this comparison for an open channel flow at $Re_\tau = 10^8$ driven by a constant body force $f_1$. The friction Reynolds number $Re_\tau = u_\tau h/\nu$ is based on the channel height $h$ and the friction velocity $u_\tau^2 = hf_1$ here.

The lower surface is characterized by a roughness length $z_0$ that results in a non-dimensional channel height of $h/z_0 = 10^4$. The solution is computed in a domain of size $2\pi h$ in streamwise and $(4/3)\pi h$ in cross-stream direction. The velocity field is



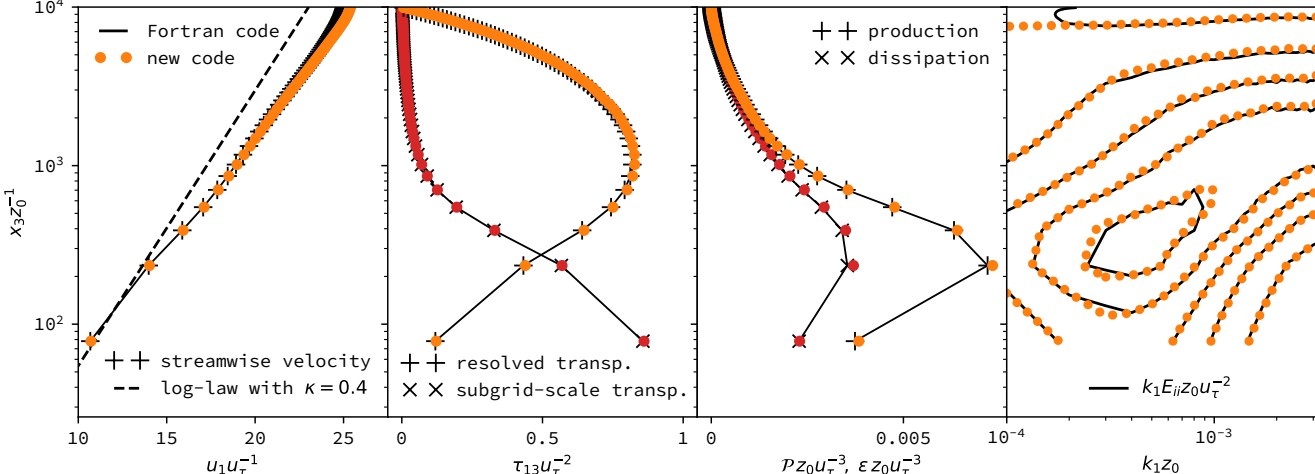

**Figure 3.** Large-eddy simulation (LES) of a turbulent channel flow at $Re_\tau = 10^8$ with an aerodynamically rough wall and a channel height of $h/z_0 = 10^4$, validated against a tried and tested Fortran code with the same numerical approach (Giometto et al., 2017). All values are non-dimensionalized with the friction velocity $u_\tau$ and the roughness length $z_0$. Mean profiles are shown for the streamwise velocity $u_1$, the resolved transport $u_1 u_3$, the subgrid-scale transport $\tau_{13}^{\mathrm{sgs}}$, as well as the production $\mathcal{P}$ and (pseudo-)dissipation $\varepsilon$ of resolved turbulent kinetic energy. The last panel shows contours of the resolved turbulent kinetic energy spectra $E_{ii}$ along the streamwise direction, premultiplied with the wavenumber $k_1 = 2\pi\kappa_1/L_1$.

discretized with $63 \times 63$ Fourier modes at $64$ equidistant vertical grid points. The mean statistics computed over $80$ large-eddy turnover times $T_\tau = h/u_\tau$ after a spin-up time of $20\,T_\tau$ closely match for the two separate implementations.

Combined, these validation efforts provide ample evidence that the implementation matches the mathematical formulation of the methods and that those methods are capable of accurately simulating flow physics, within the limitations of the physical models. A comprehensive set of easily repeatable validation tests serves both to verify the current implementation and to ensure that future developments do not jeopardize correctness. This should not only facilitate adding new functionality but also help making changes to existing functionality and avoid getting locked into design decisions that might prove suboptimal for future developments.

## 4  Performance and scaling

Defined in a broad way, performance can be understood as the time required to obtain a solution at the required quality given the available computational resources. We can examine how the time changes as a function of the required quality and the available resources (relative performance) or how fast different methods arrive at a solution for fixed quality and resources (absolute performance). However, it is difficult to measure the overall quality of a turbulent flow simulation in a quantitative way since the system is chaotic and analytic solutions are not available. Furthermore, there is a great diversity of computational





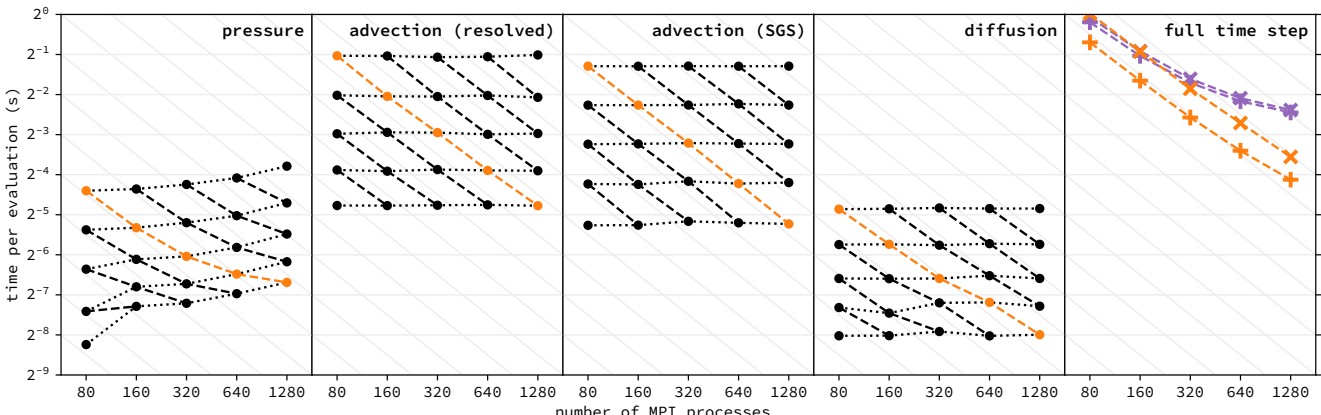

**Figure 4.** Performance and scaling of individual terms and full time step, as measured on Intel Xeon Ice Lake nodes of the Stampede2 system at the Texas Advanced Computing Center. Individual terms are computed at a resolution of $256 \times 256 \times \lambda N_p$, where $\lambda \in \{1, 2, 4, 8, 16\}$ is the number of vertical grid points per MPI process and $N_p$ is the total number of MPI processes. Dotted lines indicate weak scaling, dashed lines indicate strong scaling, and the grid lines correspond to perfect scaling. The last panel shows the overall performance for the resolution $256 \times 256 \times 1280$ (highlighted in orange on other panels) in comparison to a pre-existing Fortran implementation of the same numerical approach (Giometto et al., 2017). DNS performance is shown with the symbol $+$, LES performance with the symbol $\times$.

resources that vary along important dimensions such as floating point operations per second (FLOPS), memory bandwidth and latency, network bandwidth and latency, and many more. We therefore narrow the scope of the performance analysis to the question of the time required to obtain a solution given the specified simulation parameters and the number of compute nodes, as measured on a fairly typical high-performance computing (HPC) system.

For the implemented explicit time integration schemes, the computational cost of a single evaluation of the right-hand side of $\mathrm{d}\boldsymbol{u}/\mathrm{d}t = f(\boldsymbol{u})$ fully characterizes the overall cost of a simulation, which is a simple function of the number of steps and the evaluations per step (i.e. stages of a Runge–Kutta method). The computational cost of a single evaluation of $f(\boldsymbol{u})$ depends primarily on the number of Fourier modes and vertical grid points and whether a subgrid-scale term is modeled (LES) or not (DNS). The impact of other parameters such as the type of pressure forcing (constant-flux vs. constant-force), boundary
conditions, and grid transformations is imperceptible.

   To assess relative performance, Fig. 4 shows the time required to compute the advection, diffusion, and pressure terms for different numbers of compute nodes and vertical grid points. The figure displays both strong scaling, where the number of processes is varied for a problem of fixed size, and weak scaling, where the problem size is varied in proportion to the computational resources. These results shows that the advection term contributes most to the overall cost while the pressure
term exhibits the most problematic scaling behavior.

   Computing the advection term is a global operation in horizontal direction but only involves neighboring nodes in vertical direction. The bulk of the computational work consists of computing discrete Fourier transforms, which are local to each MPI process and scale as $\mathcal{O}(N \log N)$ where $N$ is the number of modes. It appears that this cost dominates over the cost





of communication, resulting in near-perfect strong and weak scaling. When computing subgrid-scale stresses with a static

Smagorinsky model, additional transforms are required and the cost increases to almost twice as much without affecting the scaling behavior.

Computing the pressure term has no data dependency between horizontal modes but is a sequential, global process in vertical direction (Thomas algorithm). Horizontal modes can be processed in batches to stagger the sequential passes up and down the domain, where the size those batches is a tuning parameter that represents a trade-off between maximizing parallelism

and minimizing per-batch overhead. The resulting performance shows imperfect weak and strong scaling. Scaling appears to improve when there are more vertical grid points per process, increasing the work-to-communication ratio. While the overall cost appears to remain at most about a quarter of the cost of the advection term, it is possible that the two costs are even closer for some combinations of hardware configurations and simulation parameters, in which case it could be worth optimizing the batch size parameter of the pressure solver.

Computing the diffusion term only involves neighboring vertical grid points and has no global data dependencies. This results in near-perfect weak scaling. Strong scaling is not quite perfect, which is explained by the fact that there is very little work to do for each grid point so the work-to-communication ratio is low. This has no discernible effect on the overall scaling behavior however, as computing the diffusion term is always at least an order of magnitude less work than computing the advection term.

To assess absolute performance, Fig. 4 includes a comparison with a Fortran code that implements the same numerical methods (Giometto et al., 2017). While such a comparison does not answer the question of whether either code is making optimal use of the computational resources, it does respond to the practical question of whether there are any performance trade-offs when substituting the new code for a codebase that has been actively used for turbulence research for over two decades. The comparison shows that the overall performance of both implementations is of a similar order of magnitude, with

the new Julia implementation showing somewhat better scaling and significantly faster DNS performance.

It appears that the new Julia code has avoided introducing excessive overhead without much effort devoted to performance optimization. That it even surpasses the performance of the Fortran implementation is likely explained by two factors. First, the new code is formulated with the Fourier domain representation at its center, which makes it easier to avoid unnecessary Fourier transforms than in the physical-space formulation of the Fortran implementation. Second, the new code makes different

trade-offs between work and communication which appear to be more suitable for modern hardware. Some of these insights will flow back to the Fortran implementation, reducing the performance discrepancy between the two codes.

Overall, the performance characteristics of the new code are as expected. The computational cost is dominated by the Fourier transforms necessary to compute the non-linear term while the pressure solver shows the least favorable scaling properties, and the overall performance is comparable to a Fortran implementation of an equivalent numerical scheme. The analysis shows that

for the implemented numerical scheme and current HPC hardware, performance is optimized by reducing the number and size of Fourier transforms and choosing an efficient implementation of the fast Fourier transform algorithm. Other details matter less as long as the computational cost can be kept significantly below the cost of the non-linear term.



Future improvements are likely to focus on parallelism along horizontal coordinate directions, either through multi-threaded CPU code or through GPGPU computation, allowing the code to scale to larger systems. There is also room for optimization in

how communication is handled, which might become important if the work-to-communication is decreased through additional parallelism. However, performance optimizations always have to be weighed against their impact on code simplicity and ease of adaptation. Since the computational cost of flow simulations is a strongly non-linear function of the grid resolution, large performance differences are required for a practical difference in the scientific problems that can be tackled.

### 4.1 On the suitability of Julia for high-performance computation

Conceptually, there are two main differences in the performance characteristics of Julia compared to languages like Fortran, C, and C++. The first difference concerns the handling of types. For a statically typed language like Fortran, the type of every variable is specified and therefore available to the compiler, which can use the information to generate efficient machine code. In Julia, machine code is generated when a function is called for the first time, at which point the types of the function arguments are known. For subsequent function calls with the same argument types, the compiled code is reused while a new copy of the

function is compiled for calls with different argument types. If the types of all variables inside a function can be derived from the type of function arguments, the compiler has access to the same information as for a statically typed language and can in principle generate the same or equivalent machine code. The second difference concerns the handling of memory allocations. For a language with (semi-)manual memory management like Fortran, memory is allocated and freed either manually with explicit commands or following deterministic rules. In Julia, memory is automatically allocated whenever required for the

operation that is performed and the program is periodically interrupted to examine which of the memory is still in use and which memory can be freed (garbage collection). Vectorized Julia code written in a naive way often allocates large amounts of memory for intermediate results, but writing Julia code that operates in-place and avoids such allocations is relatively easy with experience. Therefore, there is no *a priori* reason that code written in Julia should be significantly slower (or faster) than code written in Fortran, C, or C++, although Julia does make it much easier to accidentally include expensive operations that

result in poor performance.

Julia was chosen as implementation language for this code with the goal of improving the ease of development, hoping that the negative effect on performance could be kept minimal. However, the experience so far has shown that the net impact on performance might even be positive. Performance optimizations can be seen as a continuum from low-level (e.g. vectorized CPU instructions, optimal use of CPU cache) to high-level (e.g. choice of algorithms, speed–accuracy trade-offs). At the lower

end of this range, Julia relies on the LLVM compiler framework and a number of pre-existing libraries, making use of countless hours of optimization work, but Julia also makes it rather easy to write code it cannot optimize very well. Maintaining close-to-optimal performance therefore requires regular measurements and occasional fixes. High-level optimizations on the other hand require understanding the performance characteristics of different approaches and choosing the right one, often by measuring the performance of different implementations. This type of optimization work benefits from the Julia language features and the

ease of integrating packages from a growing ecosystem. The impact of language choice on performance depends not so much



on what optimizations are theoretically possible but which ones are simple enough that they are done in practice. It is therefore possible that the choice of Julia will be a benefit rather than a drawback for the performance of this code over its lifetime.

## 5 Conclusions

Turbulence-resolving ABL flow simulations are subject to a number of competing requirements that have to be considered when developing simulation code. Availability of physical and numerical models, performance and scalability, ease of use and ease of modification, safeguards against implementation and usage errors, as well as license terms may vary considerably and trade-offs are often inevitable. The Julia programming language is promising more favorable trade-offs by offering the ergonomics of a modern high-level language without sacrificing performance.

In this paper, we have introduced a new code for turbulence-resolving flow simulations, designed for the requirements of atmospheric boundary-layer research and written in Julia. The performance is shown to be in line with a Fortran implementation of the same modelling approach. In fact, it even appears that easier experimentation with algorithmic approaches and implementation trade-offs might have a stronger impact on performance than the remaining computational overhead compared to highly optimized Fortran compilers.

The code also places a focus on continuous testing and minimizing the chances for errors both during development and usage. This is particularly important in exploratory research, where the expected behavior of a new model or flow system is not known *a priori* and it is difficult to discern between inconspicuous errors and novel results.

The code provides the core functionality for both direct numerical simulation and large-eddy simulation in channel-flow geometries. In the future, we expect to expand the scope by adding functionality such as more advanced subgrid-scale models, support for temperature, humidity and transport of passive scalars, and partially resolved complex terrain.

*Code and data availability.* BoundaryLayerDynamics.jl is open source software and available under the MIT License through the official Julia package repository and on GitHub[2], where the public repository of the package is currently hosted. The version described in this article is archived on Zenodo (Schmid, 2023). The data and code required to reproduce this paper are also made available on Zenodo (Schmid et al., 2023).

*Author contributions.* MFS developed the model code, performed validation and performance testing, and prepared the manuscript. MGG, GAL, and MBP acquired project funding, supervised the work, and reviewed the manuscript. Access to computational resources was provided by MGG.

*Competing interests.* The authors declare that they have no conflict of interest.

---

[2]https://github.com/efpl-columbia/BoundaryLayerDynamics.jl



*Acknowledgements.* We acknowledge the financial support from the NSERC (Canada), the Swiss National Science Foundation, the Department of Civil Engineering and Engineering Mechanics at Columbia University, Monash University, and the University of Rhode Island. This
work made use of the Stampede2 system at the Texas Advanced Computing Center (TACC) through the Extreme Science and Engineering
Discovery Environment (XSEDE; Towns et al., 2014), which is supported by National Science Foundation grant number ACI-1548562.



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
