# Peer review of "BoundaryLayerDynamics.jl v1.0: a modern codebase for atmospheric boundary-layer simulations"

_EGUsphere, 2023_

## Author Response (AR1)

**Author's Response to Comments**

We thank the reviewers for their thoughtful comments. We have prepared a revised version of the manuscript based on these comments. Replies to individual comments have been posted to EGUsphere and are included below.

**Community Comment by Zheng Gong**

> *Typo errors exist in the Eqs.(2) and (3) for the viscous term.*

Thank you for noticing this error. The second derivatives were indeed not correctly converted to the GMD formatting, resulting in erroneous diffusion terms in Eq. (1) and (2). This will be fixed in the submission of the revised manuscript.

**Referee Comments by Michael Schlottke-Lakemper**

> ***Sec. 3***: *It is not clear which tests are run automatically, the authors remain somewhat vague. Since they spent a good part of the section on CI testing, it would be good to give an overview of all tests (or at least the main test categories) and which are run automatically.*

There is a lot of different nomenclature around software testing, but I understand "automated testing" and "continuous testing/integration" as two distinct concepts (see e.g. https://en.wikipedia.org/wiki/Test_automation and https://en.wikipedia.org/wiki/Continuous_testing). The discussion in the paper is focused on the former and there is currently no continuous integration (CI) set up. I have removed a potentially misleading use of "continuous testing" in the conclusion section for the revised manuscript.

When the paper refers to "automated tests", this is meant to convey that those tests consist of a computer program that can be run with a simple command and produces a "pass/fail/error" result. In practical terms, this means that there is a `runtests.jl` file that makes use of the `@test` and `@testset` macros to run and verify those test cases. All test cases described as "automated tests" in section 3 (model validation) are implemented in that way, i.e. all cases that do not involve turbulent flows.

The model validation section is meant to give an overview of the test categories. As explained in the first paragraph, the testing includes (a) automated testing of individual right-hand-side terms for a prescribed velocity field, (b) automated testing of the time

integration of ODEs, (c) automated testing of laminar 2D flows, and (d) non-automated tests of turbulent flow simulations.

> ***Sec. 3***: *It is not clear where and how the automated tests are run (e.g., GitHub Actions, Jenkins, and on which hardware). Also, the number of ranks used for testing is left unclear. Overall, this part of the manuscript reads more like a manual for users rather than a scientific paper. I thus recommend to either remove the more generic content or to be more specific and add details such that a reader can learn about how CI testing is implemented for the described code.*

Currently there is no continuous integration set up and tests need to be re-run by the person making or reviewing changes to the code on their own machine. With the current pace of development this has been working well enough, but we will likely set up some form of continuous integration in the future. The automated tests do not include any performance measurements so they should be independent of the hardware on which the tests are run.

The number of ranks in the automated tests is set to 4 by default, as this includes the special cases of the lowest and highest process and can be run easily on 4-core CPUs that are common in consumer hardware. However, that number is a parameter of the `runtests.jl` script and can be changed easily to test other process counts.

I think that the impression of a manual-like tone is mostly due to the fourth paragraph in the validation section. I have reformulated this paragraph in the revised manuscript.

> ***Sec. 3, lines 236 ff.***: *Are tests included that verify that the result of a simulation does not change with the number of MPI ranks? Either way, it would be good if the authors would comment on that since using different numbers of ranks to verify identical results is a commonly used practice in codes that support it.*

The tests currently do not perform any direct comparison between results that are computed with different numbers of ranks. However, since the tests perform comparisons with analytical solutions, they do verify that the code produces the same result independent of the number of ranks.

> *The reproducibility data was clearly curated in a git repository (inferred from the presence of git-specific files). However, it is only available as a zip downloaded from Zenodo. This is both not very user friendly (there is no online code browsing) and makes it harder to find then necessary (ref. the F in FAIR). It would be good if the reproducibility data was available as, e.g., a public GitHub repository that is linked to from the DOI.*

Relying on a proprietary commercial platform (that is regularly censored in parts of the world) as an integral part of open publishing does not seem ideal to me, but I do

recognize that the user experience for browsing files on Zenodo is rather lacking. I am not sure whether findability is affected, as a permanent DOI link seems more reliable than a GitHub link that can change and disappear at any time, and if the GitHub repository is linked from Zenodo, the data has already been found. However, having a copy of the data and code on GitHub should generally make it easier to access and reuse the files. I have therefore pushed a copy of the repository to https://github.com/mfsch/paper-boundarylayers.jl-v1.0 and linked this page from the Zenodo entry.

> *The reproducibility data does not give any explanations on how to use it to reproduce the results in the manuscript. At the very least a README.md should be included that describes the contents of the data collection, states the Julia version that has/can be used to obtain the results in the paper, and information on how to re-run the experiments, post-process the data etc. As it is now, even though the reproducibility data seems to be fairly exhaustive, it is hard to use it without extensive effort (ref. the R in FAIR).*

I have added a section to the main README that gives additional information on how the repository is organized and how different steps can be rerun. The Makefiles include the exact commands that are needed to run each step.

> *p. 2, line 34: I think the reference to the new package BoundaryLayerDynamics.jl (Schmid 2023) should appear at its first use in this line.*

I have added the reference in the revised manuscript.

**Referee Comments by Cedrick Ansorge**

> *The formulation of the vertical grid, evaluating the function $x_3$ on the center point in terms of $\zeta$ introduces additional truncation errors that depend on $dx_3/d\zeta$. (Values are approximated as arithmetic means – for instance in Eq. (10) or line 137), but for large stretching, this is not exact). These errors should be taken into account when the order of accuracy is discussed and they will result in a constraint on the grid mapping function $x_3(\zeta)$ that limits stretching. This is of particular relevance for the choice of grid taken here with $b\,\eta = 0.97$ which for n=96 yields stretching from grid point to grid point of more than 50%.*

The truncation error always has a prefactor that depends on derivatives of the approximated function, e.g. $df/dx_3$. If we apply a coordinate transform, this prefactor becomes $df/d\zeta$, which can also be expressed as $dx_3/d\zeta\,df/dx_3$. The order of convergence is unaffected, as these factors do not change with the grid spacing. Whether the prefactor increases or decreases the truncation error depends on the functional form of both $f(x_3)$ and $x_3(\zeta)x$ and cannot be stated in general. Indeed, the goal of grid stretching is generally to select a coordinate transform that reduces the maximum value of this prefactor.

*Typo in Eq. 1 (need second derivative for viscous term)*

The second derivatives in Eq. 1 and 2 have been fixed.

*l. 110 / Eq. (8): How can the model include an approximate formulation for $\tau$?*

Eq. (3) & (8) were using the "approximately equal" sign to indicate that the relation is a modeling assumption rather than a mathematical identity. I have now replaced those with the regular "equal" sign to avoid confusion, along with a number of other potentially confusing occurrences of the "approximately equal" sign.

*l. 147: The commutation of operators precludes their linearity; the truncation is, however, non-linear such that, in their discrete representation, the operators do not commute. The choice might impact on stability and conditioning properties of the algorithm and needs to be motivated / detailed here.*

I am not sure if I understood the reviewer's concern correctly, especially the statement "the commutation of operators precludes their linearity", but I hope the following explanations address any doubts.

The commutativity properties we rely on apply to the discrete operators and are exact to floating-point precision. Computing vertical derivatives in the physical domain as opposed to the frequency domain can sometimes be used to reduce the number of necessary FFTs, but it should not affect the results any more than the other ways floating-point math is inexact (e.g. non-associativity of addition).

The claim that vertical derivatives commute with horizontal FFTs (forward and backward) corresponds to the Julia expression `rfft(diff(x, dims = 3), (1, 2))` $\approx$ `diff(rfft(x, (1, 2)), dims = 3)`, which can be verified with e.g. `x = rand(8, 8, 8)` (and similarly for inverse transforms). If there are non-zero contributions from the boundary conditions, these have to be transformed as well of course, but they should not affect the commutativity of the operations.

If the truncation that the reviewer is concerned about is when the high-wavenumber contributions are discarded, we do actually rely on the fact that this operation also commutes with the vertical derivatives, although we could just as easily always do the vertical derivatives before the truncation. This commutativity is also exact and does not even have floating-point errors (e.g. `diff(x[1:3,1:3,:], dims = 3) == diff(x, dims = 3)[1:3,1:3,:]` for `x = rfft(rand(8, 8, 8), (1, 2))`). It also happens to be a linear operation, although that alone neither guarantees nor precludes commutativity. Both linear and non-linear operations can be commutative (trivially with themselves or with the identity function) or non-commutative (e.g. two matrix–vector multiplications with random matrix coefficients).

*How is the advection discretized? Directly, in flux-form, or skew-symmetric? Has Energy conservation been checked?*

The advection term is discretized in the rotational form that is used throughout the paper (see Eq. 1, 2, and 5). This form of the advection term, in combination with the spatial discretization we use, conserves kinetic energy to machine precision as long as the products are computed with 3/2-dealiasing and there is no grid stretching. In the absence of viscosity, any change in kinetic energy is therefore due to time-differencing errors. Below is a figure that shows that this error does indeed follow the order of the time-differencing scheme, except for the SSPRK22 method that happens to be third-order in this case.

[Figure]

In flow simulations, energy conservation is subject to numerical errors (from time differencing and grid stretching), but those errors should be small enough that they do not materially affect the results. While we do not show the full energy budget for the validation cases, we show that the production and dissipation terms as well as energy spectra closely match the reference solution. We can therefore be fairly confident that the energy dynamics are reproduced correctly by the simulations.

> *Fig.4: The new implementation has a few percent (~5%?) larger mean velocity in the wake region which is quite a bit for a technical test. Is there a reason for this deviation? More or less dissipative numerics? Different grid / vertical resolution? Would these differences drop when an identical set-up was used in terms of the horizontal and vertical resolution?*

The mismatch in the upper part of the LES domain can be attributed to incomplete convergence of flow statistics. I have updated Fig. (4) with a 2.5× longer simulation run for a better match of the profiles in the revised manuscript. The numerical methods and grid resolutions are the same for both simulations.